# Effects of a New Multicomponent Nutritional Supplement on Muscle Mass and Physical Performance in Adult and Old Patients Recovered from COVID-19: A Pilot Observational Case–Control Study

**DOI:** 10.3390/nu14112316

**Published:** 2022-05-31

**Authors:** Francesco Landi, Anna Maria Martone, Francesca Ciciarello, Vincenzo Galluzzo, Giulia Savera, Riccardo Calvani, Anna Picca, Emanuele Marzetti, Matteo Tosato

**Affiliations:** Fondazione Policlinico Universitario “Agostino Gemelli” IRCCS, 00168 Rome, Italy; annamaria.martone@policlinicogemelli.it (A.M.M.); francesca.ciciarello@policlinicogemelli.it (F.C.); vincenzo.galluzzo@policlinicogemelli.it (V.G.); giulia.savera@policlinicogemelli.it (G.S.); riccardo.calvani@policlinicogemelli.it (R.C.); anna.picca@policlinicogemelli.it (A.P.); emanuele.marzetti@policlinicogemelli.it (E.M.); matteo.tosato@policlinicogemelli.it (M.T.)

**Keywords:** COVID-19, personalized medicine, frailty, geriatric syndrome

## Abstract

Objective: The purpose of the present study was to assess the effect of a specific oral nutritional supplement among patients recovered from COVID-19 but suffering symptoms of fatigue. Methods: This is an observational case–control study involving a sample of 66 COVID-19 survivors divided in two groups, 33 subjects in the intervention group who received the nutritional supplement and 33 subjects in the control group. The nutritional supplement received by subjects in the active group was based on amino acids; vitamin B6 and B1; and malic, succinic and citric acids. After an 8-week follow-up, the main outcomes considered were skeletal muscle index (measured by bioelectrical impedance analysis), physical performance measures (handgrip strength, one-minute chair–stand test, six-minute walking test), and quality of life (using EuroQol visual analogue scale). Results: All the considered areas increased significantly in the subjects receiving the active treatment with oral nutritional supplement in comparison with the baseline values. After adjusting for age, gender, and baseline values, skeletal muscle index, handgrip strength test, the one-minute chair–stand test, and six-minute walking test values were higher among participants in the treatment group compared with subjects in control group. The oral nutritional supplement significantly improved the handgrip strength; similarly, participants in the active group showed a higher improvement in skeletal muscle index, the one-minute chair–stand test, the six-minute walking test, and in quality of life. Conclusion: The nutritional supplement containing nine essential amino acids plus cysteine; vitamin B6 and B1; and malic, succinic and citric acids had a positive effect on nutritional status, functional recovery, and quality of life in COVID-19 survivors still suffering from fatigue. Additional controlled clinical trials are required to corroborate these results.

## 1. Introduction

Nutrition management and treatment is critical to improving immune responses against RNA viral infection. In COVID-19, nutritional status is an essential element across all disease phases from acute to chronic stage [1], particularly in patients at risk for adverse outcomes, such as frail, older subjects and those with multimorbidity [2]. It is widely known that malnutrition is, simultaneously, a cause and consequence of immune dysfunction. In particular, there is sufficient evidence to demonstrate that immune response can be weakened by inadequate nutrition. Furthermore, regarding COVID-19, low levels of nutritional status biomarkers (such as albumin, pre-albumin, lymphocyte) are correlated with poorer outcomes [3]. Extended hospital stays, especially in the intensive care department, is a well-recognized risk factor for potential malnutrition and correlates with a substantial deterioration in muscle mass, muscle strength, and physical performance [4].

It is clearly described that following SARS-CoV-2 infection, the high level of inflammation may exacerbate the catabolic processes and facilitate the onset of anorexia (i.e., loss of appetite and less food intake), worsening malnutrition and at the same time correlating with the longest recovery times, impaired physical performance, and reduced quality of life [5]. In this respect, the European Society for Clinical Nutrition and Metabolism (ESPEN) recommended a specific guideline for the nutritional management of subjects with SARS-CoV-2 infection [6]. According to these recommendations, nutritional screening, assessment, and therapy should be always addressed as a primary part of the continuum of care for COVID-19 subjects.

Following the ESPEN guidelines, we realized that our post-acute care service dedicated to COVID-19 survivors required a specific comprehensive nutritional assessment [7]. In particular, body composition changes (evaluated by bioelectrical impedance analysis), muscle strength modifications, as well as physical performance were evaluated, and specific nutritional recommendations were offered to help the recovery of post-acute COVID-19 patients. In particular, the objective of the present study was to assess the effect of a specific oral nutritional supplement (comprising nine essential amino acids plus cysteine, malic, succinic, and citric acids) among patients who had recovered from COVID-19 but were still distressed by symptoms of fatigue.

## 2. Methods

The Gemelli Against COVID-19 Post-Acute Care (GAC19-PAC) project is an initiative established by the Department of Geriatrics, Neuroscience and Orthopedics of the Catholic University of the Sacred Heart (Rome, Italy) with the objective to answer to the clinical and psychological needs of COVID-19 survivors [7]. Immediately after the first outbreak of the COVID-19 pandemic (April 2020), the Fondazione Policlinico Universitario A. Gemelli IRCSS set up the outpatient service named “Day Hospital Post-COVID-19” for subjects who had recovered from the SARS-CoV-2 infection. The complete GAC19-PAC study protocol has been previously addressed [7].

### 2.1. Ethical Approval and Manuscript Preparation

The Catholic University/Fondazione Policlinico Gemelli IRCCS Institutional Ethics Committee approved the entire GAC19-PAC study protocol [7]. Written informed consent has been obtained from all the patients who asked to be followed in this outpatient service. The manuscript was prepared in accordance with the Strengthening the Reporting of Observational studies in Epidemiology (STROBE) statement, describing guidelines for observational studies. Furthermore, it is important to highlight that the funding sources were not involved in the study processes (patient selection, data analyses, and results interpretation).

### 2.2. Study Sample

The Gemelli COVID-19 outpatient clinic was opened on 21 April 2020 and it is still operational. The patients who met the WHO criteria for withdrawal of quarantine were admitted to the follow-up study project [8]. For the present study, we initially enrolled 33 subjects (14 females and 19 males) suffering from fatigue at the time of baseline visit and who had received the prescription for a multicomponent oral nutritional supplement. This prescription of the nutritional supplement was part of clinical routine and it depended on the decision of the medical doctor and the nutritionist who assessed the patient. Fatigue is a well-known symptom of long COVID-19, and it is not the same as normal feelings of being tired or sleepy. Fatigue related to SARS-CoV-2 infection has been defined as a type of extreme tiredness or feeling ‘wiped out’, which persists despite being resting or getting a good night’s sleep. Fatigue related to long COVID-19 occurs even after small tasks and sometimes limits usual daily activities (i.e., difficulties in walking upstairs, doing normal tasks, or even getting out of bed).

To evaluate the effect of the nutritional supplement, 33 other subjects, matched by age and gender and who suffered from fatigue and had not received any nutritional supplement, were randomly extracted from the post COVID-19 Day Hospital database. Finally, we considered a sample of 66 subjects divided into two groups, 33 subjects received the nutritional supplement in the intervention group and 33 subjects did not and were the control group.

### 2.3. Data Collection

All subjects enrolled in the post COVID-19 Day Hospital underwent individual and multidimensional assessments [7]. Demographic information, medical and drugs history, laboratory findings, and radiological features were collected. A multidisciplinary approach, including nutritional assessment, was organized to guarantee a comprehensive assessment of all the potential damages correlated to the SARS-CoV-2 infection. During the baseline visit the health care professionals collected information and data about the persistence of signs and symptoms COVID-19 related: fatigue, cough, diarrhoea, headache, smell disorders, dysgeusia, red eyes, joint pain, shortness of breath, loss of appetite, sore throat, and rhinitis.

Body weight was assessed through an analogue medical scale. Body height was assessed utilizing a standard stadiometer. Body mass index (BMI) was calculated as weight (kilograms) divided by the square of height (meters).

Skeletal muscle mass was assessed by bioelectrical impedance analysis (BIA). Bioelectrical impedance analysis resistance (ohms, W) was achieved utilizing the InBody machine [9]. Direct segmental measurement bioelectrical impedance analysis regards the human body as five cylinders: left arm, right arm, torso, left leg, and right leg. InBody independently measures each cylinder to provide accurate measurements for the entire body. With InBody’s direct segmental technology and 8-point tactile electrodes it is possible to obtain more accurate body composition results. Subjects were asked to void shortly prior to the BIA evaluation since the impedance analysis and results are the function of the resistance of electrical current against water flow. No other specific recommendations were given.

Muscle strength was assessed by handgrip strength testing using a North Coast hand-held hydraulic dynamometer (North Coast Medical, Inc., Morgan Hill, CA, USA) [10]. The participant was seated on a chair with shoulder in a neutral position, the elbow near the trunk and flexed at 90°, and the wrist in a neutral position (thumbs up). After one familiarization trial, muscle strength was assessed in both hands and the highest value (kg) was considered.

Physical performance was assessed by the one-minute sit-to-stand test and the six-minutes walking test. These tests are usually utilized to evaluate exercise-induced respiratory impairment in subjects with pulmonary illnesses [11]. For the sit-to-stand test, participants were asked to stand up from a chair and sit down with their arms folded across the chest for one minute as quickly as possible. The number of repetitions was recorded, and the higher numbers reflect the better performance [12]. The six-minutes walking test was performed on a 20 m long track and the distance covered (meters) was recorded; a longer distance covered reflects a better performance [13].

Finally, the EuroQol visual analogue scale (VAS) was used to obtain a rapid evaluation of self-rated health, on a scale from 0 to 100, with 0 indicating the worst imaginable health and 100 corresponding to the best imaginable health [14].

### 2.4. Intervention—Oral Nutritional Supplement

Thirty-three patients received 8-week treatment with a multicomponent nutritional supplement (food for special medical purposes) based on ten amino acids and malic, succinic, and citric acids (Amino-Ther Pro—Professional Dietetics, Milano, Italy). The composition of each portion is described in Table 1. Patients were recommended to have two portions a day away from principal meals (for example, mid-morning and mid-afternoon).

### 2.5. Statistical Analyses

Sample power was calculated based on physical performance measures, considering a potential power of 80% and an α-error of 5% with an estimated positive result among subjects taking the nutritional supplement of 90%, related to 40% of the control subjects. Based on this evaluation, 60 participants were required to have an 80% probability of observing a positive result for the primary outcomes, from 90% in the subjects treated with the multicomponent nutritional supplement to 40% in the participants in the control groups.

Continuous variables were reported as mean ± standard deviation (SD), categorical variables as frequencies by absolute value and percentage (%) of the total. Descriptive statistics were utilized to define demographic and principal clinical characteristics of the study sample according to the treatment. The differences in proportions and the means of all the variables considered were evaluated by means of Fisher’s exact test and t test statistics, respectively.

Finally, the analysis of covariance (ANCOVA) adjusted for age, gender, and baseline values was used to evaluate the result of oral multicomponent nutritional intervention on skeletal muscle mass, physical performance measures, and quality of life at the end of an 8-week observational period.

Analyses were executed using SPSS software (version 11.0, SPSS Inc., Chicago, IL, USA).

## 3. Results

The mean age of the 66 subjects participating the study was 61.0 years (standard deviation 11.8, range from 39 to 75 years), and 29 (44%) were women. Characteristics of the study population according to the treatments are summarized in Table 2. Overall, subjects in the treatment group had similar acute COVID-19 severity compared to participants in control group, considering the similar hospitalization rate, percentage of oxygen therapy, and non-invasive and invasive ventilation. For the subjects who needed hospitalization, the length of hospital stay was similar between intervention and control group as well.

About three months after the onset of the acute disease, body mass index was significantly lower among subjects in the treatment group compared to those in the control group (23.9 ± 3.3 versus 25.8 ± 3.8, respectively; *p* = 0.03). Even though the scores of skeletal muscle mass, physical performance measures, and quality of life were marginally higher among subjects in the control group compared to the intervention group, there were no significant differences between the groups of interest.

In addition to fatigue, other COVID-19 related persistent symptoms were present without significant differences (Figure 1). Persistent joint pain was the only symptom more frequently observed among subjects in the intervention group compared to those in the control group (36% versus 18%, respectively; *p* = 0.01).

None of the subjects in the active group stated any difficulty in taking the multicomponent nutritional supplement concerning volume, palatability, and taste. The treatment lasted on average was about 8 weeks; treatment compliance was good (1.5 portions per day) and no side effects were reported.

The mean values of skeletal muscle index, physical performance measures, and quality of life after an 8-week observation are shown in Table 3. All these domains significantly increased in comparison with baseline values in the subjects receiving the active treatment with the oral nutritional supplement. After adjusting for age, gender, and baseline values, skeletal muscle index, handgrip strength test, the one-minute chair–stand test, and the six-minute walking test values were higher among participants in treatment group compared with subjects in the control group.

Finally, for these measurements, we considered the percentage change between the follow-up values versus the baseline values in both groups of interest (Figure 2). Notably, oral nutritional supplement significantly improved the handgrip strength more than observed among subjects in the control group; similarly, participants in the active group showed a higher improvement in skeletal muscle index, the one-minute chair–stand test, the six-minute walking test, and in quality of life.

## 4. Discussion

International guidelines and expert consensus recommend appropriate and timely nutritional evaluation and specific approaches to improve clinical outcomes in subjects at higher risk of developing nutritional problems, including subjects with chronic and/or severe diseases in different healthcare settings [6,15,16]. The present study suggests and confirms the importance of nutritional supplementation in subjects suffering from an infection and catabolic disease, such as COVID-19. A multicomponent nutritional supplement, containing nine essential amino acids plus cysteine, malic, succinic, and citric acid, and vitamin B6 and B1, supported the improvement of skeletal muscle index and the increase of physical performance among subjects who had recovered from COVID-19 but were still suffering from persistent fatigue. Furthermore, nutritional support contributed to the improvement of a better quality of life.

The nutritional status of COVID-19 patients should be part of the multidimensional assessment, especially at an early stage of disease [17]. Subjects with SARS-CoV-2 infection and nutritional problems typically have a high inflammatory response, which may represent the biological substrate of the malnutrition [18]. In addition, malnutrition is significantly associated with poor outcomes of COVID-19, while the prognosis of subjects without malnutrition is relatively favourable [19].

It has been clearly established that even twelve weeks after the onset of COVID-19, more than 50% of patients continued to suffer from fatigue and 15–20% still had smell disorders, distorted taste, and loss of appetite [5,19]. With the persistence of these conditions, it is important to advocate for specific nutritional assessment protocols and the prescription of multicomponent nutritional support. The estimation of energy, protein, special micronutrient, and oral nutritional supplement needs to be implemented in accordance with the appropriate guidelines in both non-COVID-19 and COVID-19 patients, while taking into consideration all the specific requirements of the older persons and/or polymorbid patients [20,21].

The nutritional supplement (mainly based on ten amino acids; vitamin B6 and B1; and malic, succinic, and citric acids), dispensed to subjects in this pilot study, was shown to help patients meet the nutritional requirements of post-COVID-19 recovery and in enhanced not only the nutritional domains, such as skeletal mass index, but also the physical performance and quality of life.

The pharmaceutical formula containing a combination of essential amino acids and some intermediate substrates of the tricarboxylic acid cycle of the mitochondria can ensure a more intense activity of protein synthesis, mitochondrial-genesis, and mitochondrial activity. The amino acids play an important role due to their major mediating effects on protein synthesis, glucose homeostasis, weight gain, and stimulation of signalling pathways (e.g., phosphoinositide 3-kinase-protein kinase B, the mammalian target of rapamycin) [22]. Researchers have also focused on the effects of amino acids on immune function, both directly and through modifications of the intestinal microbiota. In particular, amino acids can be incorporated into and oxidized by immune cells [23], supporting immune cell function [24]. Furthermore, considering the mechanism of SARS-CoV-2 transmission, it also important to highlight that amino acids can improve mucosal surface defense by stimulating IgA secretion [25]. Amino acids may modulate immune response, enhancing immune cell energy sources, the CD4+, CD4+/CD8+ ratio, intestinal immunoglobulin, innate and adaptive immune response, pro-inflammatory cytokines, and dendritic cell function [26,27]. Even though we have no measurements regarding the composition of the intestinal microbiome, in light of the correlation between the amino acid pool and the proper functioning of the immune system, it is reasonable to suggest the achievement of the correct amino acid intake, also through oral supplementation, in COVID-19 survivors.

The other important component of the oral supplement used in the present study is the combination of malic, citric, and succinic acids. These acids are organic compounds (malic and succinic acids are dicarboxylic acids, while citric acid is a tricarboxylic acid), playing an important role in biochemistry, especially in the citric acid cycle, also known as the TCA (tricarboxylic acid cycle) or Krebs cycle [28]. In eukaryotic cells, the citric acid cycle occurs in the matrix of the mitochondria. It consists of a series of redox reactions [28]. The reduced electron carriers generated in the TCA cycle transfer electrons to electron transport chain and produce most of the ATP that is generated in cellular respiration. In addition to its function in energy-yielding metabolism and the oxidation of 2-carbon units, the TCA cycle is also one of the most important pathways for inter-conversion of 4- and 5-carbon compounds in the cell, many of which occur from amino acids or act as intermediates in the synthesis of them. Some studies have demonstrated that the combination of a cluster of amino acids and three organic acid of the TCA cycle (malic acid, succinic acid, and citric acid) is crucial for protein synthesis. The precise stoichiometric ratio of amino acids and these three organic acids is able to act as a mitochondrial metabolism modulator, enhancing mitochondrial energy production and aiding protein metabolism.

Overall, considering this biological plausibility, oral supplements containing amino acids, succinic acid, malic acid, citric acid, and vitamins B1 and B6 could be indicated in cases of low intake through diet or increased need, such as in older subjects affected by COVID-19 [29]. Furthermore, considering that long COVID-19 syndrome is characterized by an increase in inflammatory markers, the combination of these three organic acids with amino acids could be useful for a reduction of the inflammatory status. Although demonstrating this significant result, our study protocol has numerous limitations that need to be discussed. First, this study should be considered a pilot study presenting a significant improvement of muscle mass and physical performance; nevertheless, it is critical to underline that the small number of subjects involved needs confirmation in larger samples. Second, this is a retrospective case–control observational study and consequently the results found cannot be directly related to the multicomponent nutritional support. It is reasonable that the enhancement in skeletal muscle index, physical performance, and quality of life is simply correlated to the longer time since the onset of COVID-19. Considering this study did not use a randomized, double-blinded design, test subjects were prescribed the supplement by a healthcare provider (medical doctor and/or nutritionist) after baseline evaluation. Thus, it is possible that the test subjects were in a worse condition and therefore exhibited greater improvement during the recovery period. This possibility is backed up by the data showing the higher (worse) prevalence of 10 out of 12 COVID-19-related symptoms in the test group compared to controls. Finally, the major limitation is in the potential bias due to the selection of subjects in the control group. Even if the two groups did not show significant differences, it is possible that the subjects selected for nutritional treatment had clinical characteristics that made them more at risk of greater improvement in nutritional status. At the same time, the subjects in the control group may not have received the nutritional prescription because they were assessed a priori as non-responders. Nevertheless, our data are promising and could be the base for a larger clinical trial.

## 5. Conclusions

Despite the limitations described, it is important to highlight that nutritional intervention can improve muscle mass, physical performance, and quality of life in COVID-19 adult and old survivors. In this respect, the hypothesis that the nutritional supplement based on 10 amino acids (Leucine, Lysine, Threonine, Isoleucine, Valine, Cysteine, Histidine, Phenylalanine, Methionine, Tryptophan); vitamin B6 and B1; and malic, succinic, and citric acids may have had a positive impact on nutritional status and functional recovery is corroborated by the scientific evidence and by a strong biological plausibility. Additional studies, in particular controlled clinical trials, are desirable to validate these results. However, it important to highlight that the assessment of nutritional status and the treatment with adequate multicomponent nutritional support is advised in all patients in the acute stage of COVID-19 and in the post-acute phase as well.

## Figures and Tables

**Figure 1 nutrients-14-02316-f001:**
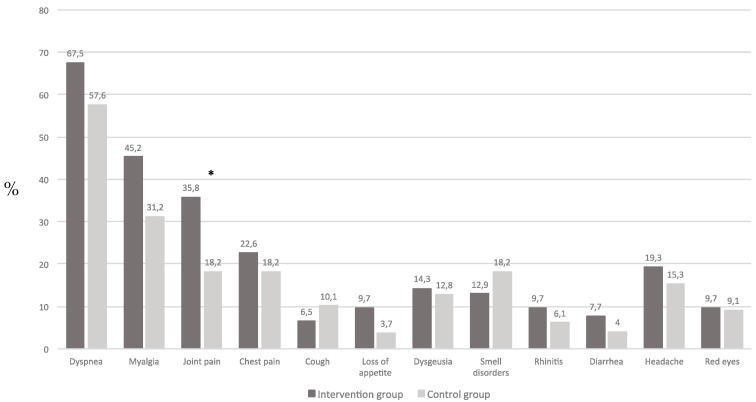
Prevalence of related COVID-19 symptoms at the time of baseline visit according to treatment (* *p* = 0.01).

**Figure 2 nutrients-14-02316-f002:**
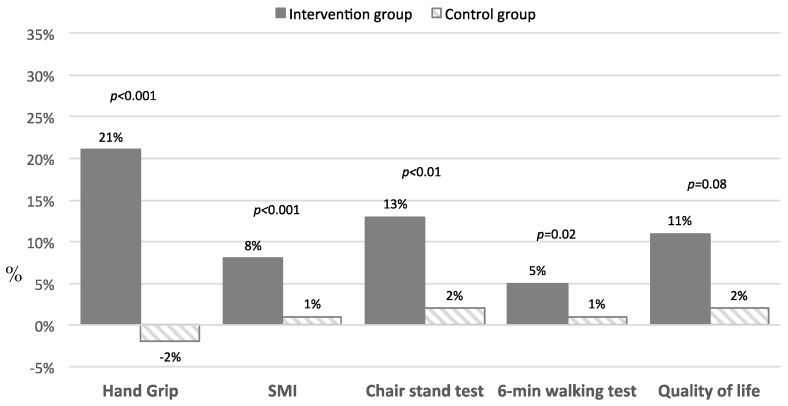
Percentage changes in skeletal muscle index, physical performance measures, and quality of life between baseline and follow-up.

**Table 1 nutrients-14-02316-t001:** Characteristics of oral nutritional supplement Amino-Ther Pro (Professional Dietetics).

Nutritional Characteristics	Amount Per Portion
Energy (kcal)	22.9
Lipids (g)	-
Carbohydrates (g)	21
Protein (g)	-
L-Leucine (mg)	1200
L-Lysine (mg)	900
L-Threonine (mg)	700
L-Isoleucine (mg)	600
L-Valine (mg)	600
L-Cysteine (mg)	150
L-Histidine (mg)	150
L-Phenylalanine (mg)	100
L-Methionine (mg)	50
L-Tryptophan (mg)	50
Vitamin B6 (mg)	0.85
Vitamin B1 (mg)	0.70
Citric acid (mg)	409
Succinic acid (mg)	102.5
Malic acid (mg)	102.5

**Table 2 nutrients-14-02316-t002:** Characteristics of study population according to intervention *.

Characteristics	Total Sample (*n* = 66)	Intervention (*n* = 33)	Control (*n* = 33)	*p*
Age (years)	61.0 ± 11.8	61.2 ± 11.1	60.8 ± 12.7	0.88
Gender (female)	29 (44)	14 (43)	15 (46)	0.50
Acute COVID-19 characteristics				
Hospitalization	33 (50)	17 (51)	16 (48)	0.50
Pneumonia diagnosed	27 (40)	15 (43)	12 (36)	0.23
Intensive care unit admission	8 (12)	4 (12)	4 (12)	0.78
Oxygen therapy	24 (36)	14 (42)	10 (30)	0.12
Noninvasive ventilation	7 (10)	4 (12)	3 (9)	0.54
Invasive ventilation	5 (7)	3 (9)	2 (6)	0.22
Length of hospital stay	15.7 ± 10.6	16.4 ± 9.6	15.3 ± 8.8	0.36
Post-acute COVID-19 follow-up				
Days since symptom onset	94.2 ± 22.3	90.4 ± 22.5	96.8 ± 22.7	0.48
Days since negative swab	66.5 ± 19.9	57.9 ± 21.7	69.6 ± 20.1	0.21
BMI (Kg/m^2^)	24.9 ± 5.7	23.9 ± 3.3	25.8 ± 3.8	0.03
Skeletal mass index (Kg/m^2^)	7.72 ± 1.11	7.53 ± 1.04	7.91 ± 1.16	0.16
Hang grip strength (Kg)	30.8 ± 10.9	30.2 ± 11.3	31.4 ± 10.6	0.66
Chair stand test (n. repetition)	27.1 ± 10.5	24.7 ± 7.3	29.7 ± 12.8	0.06
Six-minute walking test (meter)	535.4 ± 86.2	519.8 ± 90.9	553.1 ± 78.7	0.18
Quality of life (EuroQol scale)	67.1 ± 13.5	63.8 ± 15.4	71.4 ± 10.1	0.15
Albumin (g/dL)	4.12 ± 0.33	3.79 ± 0.27	4.21 ± 0.21	0.15
Vitamin D (ng/dL)	23.7 ± 10.2	20.1 ± 5.2	25.0 ± 10.7	0.27

* Data are given as number (percent) for gender and acute COVID-19 characteristics; for all the other variables, means ± SD are reported. BMI: Body mass index; Quality of life was assessed using EuroQol visual analogue scale, ranging from 0 (worst imaginable health) to 100 (best imaginable health).

**Table 3 nutrients-14-02316-t003:** Unadjusted and adjusted means of skeletal muscle index and physical performances measures (dependent variables) according to treatment after 8 weeks of observation.

Characteristics	Intervention	Control	*p*
	(*n* = 33)	(*n* = 33)	
**Skeletal mass index** (Kg/m^2^)			
Unadjusted	8.02 ± 0.91	7.91 ± 1.16	0.6
Adjusted	8.21 ± 0.03	7.83 ± 0.03	<0.001
**Handgrip strength** (Kg)			
Unadjusted	33.2 ± 1.5	30.9 ± 1.7	0.3
Adjusted	33.7 ± 0.5	30.4 ± 0.5	<0.001
**Chair–stand test** (n. repetition)			
Unadjusted	27.9 ± 1.0	29.8 ± 1.3	0.43
Adjusted	30.0 ± 0.4	28.4 ± 0.4	0.03
**Six-minute walking test** (meter)			
Unadjusted	541.2 ± 14.3	557.7 ± 15.5	0.42
Adjusted	554.0 ± 3.1	543.1 ± 3.3	0.02
**Quality of life (EuroQol scale)**			
Unadjusted	66.4 ± 2.6	72.5 ± 2.2	0.08
Adjusted	70.0 ± 0.6	69.0 ± 0.6	0.27

Data are gives as means ± SE. ANCOVA: analysis adjusted for age, gender, and baseline value.

## Data Availability

The datasets used and/or analyzed during the current study are available from the corresponding author on reasonable request.

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
