# Peer review of "Effects of a New Multicomponent Nutritional Supplement on Muscle Mass and Physical Performance in Adult and Old Patients Recovered from COVID-19: A Pilot Observational Case–Control Study"

_nutrients, 2022, doi:10.3390/nu14112316_

Round 1

Reviewer 1 Report

Landi et al. report on a small trial (66 individuals; 33 test subjects compared to 33 age- and sex-matched controls, with the total number of subjects validated by a power analysis) to test the effects of a dietary supplement on COVID-19 survivors who were experiencing fatigue. The authors state that the study was approved by the institutional review board and written consent was obtained from the subjects. The supplement contained half of the twenty biologically relevant amino acids plus three metabolically important, 4-6 carbon organic acids (malic, succinic and citric acids). The authors conclude that the supplement is beneficial for recovery from COVID-19 because, at the end of the 8-week trial, the test group showed greater improvements in several functional parameters.

Major

The study does not use a randomized, double-blinded design. Test subjects were prescribed the supplement by their healthcare provider after evaluation. Thus it is possible that the test subjects were in worse condition and could exhibit greater improvement during the recovery period. This possibility is backed up by the data in Fig. 1 that show higher (worse) prevalence of 10 of 12 symptoms in the test group compared to controls. That said, the authors have strived to match the two populations with age- and sex-matching, and most of the characteristics in Table 2 are not significantly different. On first glance, the longer times since symptom onset and subsequent negative swab for the control group also suggest that they could be further along toward recovery, but the large variability in the two samples results in the differences being not statistically significant. Given the wide variabilities, perhaps regression analyses could provide insights into whether functional recovery is associated with any of the variables in Table 2.

The amino acids in the supplement are described as “branched-chain” but the 10 amino acids in the supplement are not all branched-chain (most are not) but include the branched-chain amino acids. Perhaps the selection of amino acids is more aptly described as the nine essential amino acids plus cysteine.

The authors have reasonably hypothesized that the supplements may be beneficial for immune function, possibly due to the intestinal microbiome. However, no measurements are provided about the composition of the intestinal microbiome, nutrient levels and WBCs in the circulation, or immune function.

Table 3 is neither described nor referred to in the text. This is curious because comparisons of four of five variables listed in the table change from not significant to significant when adjusted for age, gender and baseline value. Unless this reviewer has missed something, there are no methodological descriptions for these adjustments, and there is no discussion of the contents of the table anywhere in the manuscript.

Lines 240-242: “The nutritional supplement…was shown to be effective in meeting the nutritional needs of post COVID-19…” seems to be an overstatement relative to what was actually tested in this study.

Lines 245-248: “We have chosen this specific pharmaceutical formula containing a combination of essential amino acids and some intermediate substrates of the tricarboxylic acid cycle of the mitochondria because these two groups of substrates are enhanced ensuring a more intense activity of protein synthesis, mitochondrial-genesis and mitochondrial activity.” also seems to be an overstatement relative to what was actually measured in this study.

Despite the authors’ assertion that the nutritional supplement was well tolerated, this reviewer has to wonder if the supplement formulated to reduce any potential for digestive system irritability due to the acids? On a separate note, could the oral supplement be formulated to be safely and comfortably supplied IV instead or orally, e.g., for patients at earlier stages who are experiencing nausea?

There is a section labeled Discussion starting at line 212, and another section labeled Discussion starting on line 306. Perhaps the latter was supposed to be labeled Summary.

Lines 314-317: It seems an overstatement to say that both “…EVALUATION of nutritional status and the PRESCRIPTION of adequate nutritional support…is mandatory…” [emphasis added]  It may be more appropriate to say that they “…are advised…” [note plural, not singular] It would be premature to make a strong recommendation based on these data.

Minor

The manuscript is generally well written but there are several instances where the writing could be more precise. A few instances are given below but this is not intended to be a comprehensive list of grammatical edits.

Line 83: “…it is important highlight…” should probably be “…it is important to highlight…”

Line 106: “…short of breath…” in this context should probably be “…shortness of breath…”

Line 117: “…to obtain a more accurate body composition results.” should be “…to obtain more accurate body composition results.”

Line 129: “…higher number reflect…” should be “…higher numbers reflect…”

Line 131 “...(meter)…” should be “...(meters)…”

Fig. 1: vertical axis (%) is not labeled.

Line 225: Clarify “Nutritional status of COVID-19 patients suffered should be…”

Line 266: “In the light of…” should be “In light of…”

Line 276: “…spend their electrons in…” could be replaced with “…transfer electrons to…”  In addition, the entire sentence could be expanded to more specifically associate ETC function with generation of a proton electrochemical gradient that provides the energy for chemiosmotic synthesis of ATP.

Author Response

  • The study does not use a randomized, double-blinded design. Test subjects were prescribed the supplement by their healthcare provider after evaluation. Thus, it is possible that the test subjects were in worse condition and could exhibit greater improvement during the recovery period. This possibility is backed up by the data in Fig. 1 that show higher (worse) prevalence of 10 of 12 symptoms in the test group compared to controls. That said, the authors have strived to match the two populations with age- and sex-matching, and most of the characteristics in Table 2 are not significantly different.

We sincerely thank the reviewer for this and the following suggestions. Even though this issue was just addressed as the first important limitation of the study, in the present revised version of the paper we better describe the type of the study and all the potential correlated biases. Accordingly, new sentences clearly addressing the limitation of the study have been added in the discussion section. (Page 9, 2nd paragraph)

  • On first glance, the longer times since symptom onset and subsequent negative swab for the control group also suggest that they could be further along toward recovery, but the large variability in the two samples results in the differences being not statistically significant. Given the wide variabilities, perhaps regression analyses could provide insights into whether functional recovery is associated with any of the variables in Table 2.

Considering that there are no significant differences between the two groups (Table 2), we believe that the ANCOVA analysis adjusted for age, sex and baseline value of the independent variable is the most appropriate. A regression analysis performed only among the subjects treated with the supplement should not add useful information for the readers. However, if the Editor believes it necessary, we are ready to do it.

  • The amino acids in the supplement are described as “branched-chain” but the 10 amino acids in the supplement are not all branched-chain (most are not) but include the branched-chain amino acids. Perhaps the selection of amino acids is more aptly described as the nine essential amino acids plus cysteine.

Accordingly, the text has been modified and “branched-chain” has been deleted.(Abstract, Page 2, 3rd paragraph)

  • The authors have reasonably hypothesized that the supplements may be beneficial for immune function, possibly due to the intestinal microbiome. However, no measurements are provided about the composition of the intestinal microbiome, nutrient levels and WBCs in the circulation, or immune function.

We agree with the reviewer that we have no measurements of the intestinal microbiome. In the discussion section we have only hypothesized that the beneficial for immune function may be one of the plausible mechanisms explaining the positive results observed in the tested group. Accordingly, the text has been modified and some sentence have been deleted. (Page 8, 4th paragraph)

  • Table 3 is neither described nor referred to in the text. This is curious because comparisons of four of five variables listed in the table change from not significant to significant when adjusted for age, gender and baseline value. Unless this reviewer has missed something, there are no methodological descriptions for these adjustments, and there is no discussion of the contents of the table anywhere in the manuscript.

We sincerely thank the reviewer to highlight this “error”. The tables were now numbered in correct way. Table 3 is correctly indicated in the results section. The analysis of covariance (ANCOVA) adjusted for age, gender, and baseline values was used to examine the result of oral nutritional intervention on skeletal muscle mass, physical performance measures and quality of life at the end of 8-week observational period. This issue is already addressed in the methods section. Considering the difference in baseline values between the groups of interest, it is not surprising that the variation between baseline and follow-up values becomes statistically positive on ANCOVA analysis.

  • Lines 240-242: “The nutritional supplement…was shown to be effective in meeting the nutritional needs of post COVID-19…” seems to be an overstatement relative to what was actually tested in this study.

Accordingly, the text has been modified and the “effective” has been deleted.(Page 8, 3rd paragraph)

  • Lines 245-248: “We have chosen this specific pharmaceutical formula containing a combination of essential amino acids and some intermediate substrates of the tricarboxylic acid cycle of the mitochondria because these two groups of substrates are enhanced ensuring a more intense activity of protein synthesis, mitochondrial-genesis and mitochondrial activity.” also seems to be an overstatement relative to what was actually measured in this study.

This sentence has been modified describing only the potential benefits of the formula utilized without any emphasis about the results obtained. (Page 8, 4th paragraph)

  • Despite the authors’ assertion that the nutritional supplement was well tolerated, this reviewer has to wonder if the supplement formulated to reduce any potential for digestive system irritability due to the acids? On a separate note, could the oral supplement be formulated to be safely and comfortably supplied IV instead or orally, e.g., for patients at earlier stages who are experiencing nausea?

As specified in the results section, no side effects were recorded. To my knowledge, no similar IV formula exists on the market.

  • There is a section labeled Discussion starting at line 212, and another section labeled Discussion starting on line 306. Perhaps the latter was supposed to be labeled Summary.

The last section has been now labeled as Conclusion.

  • Lines 314-317: It seems an overstatement to say that both “…EVALUATION of nutritional status and the PRESCRIPTION of adequate nutritional support…is mandatory…” [emphasis added] It may be more appropriate to say that they “…are advised…” [note plural, not singular] It would be premature to make a strong recommendation based on these data.

Accordingly, the text has been modified and “mandatory” has been deleted.(Page 9, last paragraph)

  • The manuscript is generally well written but there are several instances where the writing could be more precise. A few instances are given below but this is not intended to be a comprehensive list of grammatical edits.

We thank the reviewer for all these suggestions/corrections. All the grammatical edits have been addressed. The y-axis label has been added (Figures 1 and 2).

Reviewer 2 Report

Thank you for the invitation to review “Effects of a New Combination of Medical Food on Muscle Mass and Physical Performance in Adult and Old Patients Re-covered from COVID-19: A Pilot Observational Case-Control Study” by Landi et al.

In this study, researchers have evaluated the impact of oral nutritional supplementation on physical function and muscle mass in patients who experience fatigue post-covid infection. I have a few points for clarification.

The study design involves a supplementation group and a control (non-supplementation) group. To me this is rather simplistic, surely to evaluate the impact of the specific amino acid supplement, the comparison should be against an inert supplement not the absence of any supplement. Have the authors accounted for the placebo effect?

Line 139: “Patients were advised to take two portions a day away from meals” – what does this mean?

Fatigue is a complex and diverse physiological phenomenon, how have the researchers standardised the magnitude/severity of fatigue amongst the participants?

Figure 2 lacks a y-axis title.                                              

The handgrip strength increases of 21% from baseline is striking, how does this compare to other similar nutritional intervention studies?

The discussion makes mechanistic inferences, based on the composition of the supplement. This is somewhat reasonable, however, I feel it needs to be dialled down a little – as given the purely physiological data in the manuscript, it is largely speculation.

Author Response

  • The study design involves a supplementation group and a control (non-supplementation) group. To me this is rather simplistic, surely to evaluate the impact of the specific amino acid supplement, the comparison should be against an inert supplement not the absence of any supplement. Have the authors accounted for the placebo effect?

We sincerely thank the reviewer for this and the following suggestions. Even though this issue was just addressed as the first important limitation of the study, in the present revised version of the paper we better describe the type of the study and all the potential correlated biases. Accordingly, new sentences clearly addressing the limitation of the study have been added in the discussion section. (Page 9, 2nd paragraph)

  • Line 139: “Patients were advised to take two portions a day away from meals” – what does this mean?

Two portions of ONS away from main meals (mid-morning and mid-afternoon). Accordingly, the text has been modified to better clarify this issue. (Page 3, last paragraph)

  • Fatigue is a complex and diverse physiological phenomenon, how have the researchers standardized the magnitude/severity of fatigue amongst the participants?

Fatigue is a well-known symptom of long COVID-19. As of April 21, 2020, the medical doctors and all the staff working at the post COVID-19 Day Hospital have accumulated a long experience in evaluating persistent symptoms after recovery from SARS-CoV-2 infection by also describing the prevalence of long COVID-19 for the first time (JAMA2020;324(6):603-605).

Fatigue in COVID-19 is not the same as normal feelings of being tired or sleepy. It’s a type of extreme tiredness or feeling ‘wiped out’ that persists despite resting or getting a good night's sleep. Fatigue occurs even after small tasks and sometimes limits the usual daily activities (i.e., difficult to walk upstairs, do normal tasks or even to get out of bed). Accordingly, the text has modified, and new sentence has been added in the methods section. (Page 2, 5th paragraph and Page 3, 1st paragraph)

  • Figure 2 lacks a y-axis title.                                              

The y-axis label has been added (Figures 1 and 2).

  • The handgrip strength increases of 21% from baseline is striking, how does this compare to other similar nutritional intervention studies?

The increase of 21% in the hand grip strength seems to be particularly significant; in reality it is important to consider the starting point of patients who at the time of the baseline assessment showed fatigue and many other symptoms related to COVID-19. As already indicated in the limitations of the study, the observed results may be related, at least in part, to the expected spontaneous recovery after six weeks of follow-up.

  • The discussion makes mechanistic inferences, based on the composition of the supplement. This is somewhat reasonable; however, I feel it needs to be dialed down a little – as given the purely physiological data in the manuscript, it is largely speculation.

Accordingly, the text has been modified and the discussion and conclusion sections are less “enthusiastic”. (Page 9, last paragraph)

Reviewer 3 Report

This is an interesting study regarding the effect of a nutritional supplement on body composition, quality of life and physical performance in patients who recovered from COVID-19. The results are important and of high clinical significance; however, the introduction and discussion sections need some improvement.

Title: The authors may consider replacing 'medical food' with ' multicomponent nutritional supplement'

Abstract and Discussion: the authors should avoid the repetition of the content of the examined supplement

Introduction: Please add a paragraph regarding the negative effect of Covid-19 on functional capacity, body composition, and quality of life of the affected people.

Methods: The lack of a placebo-controlled group is one of the main limitations of the study and should be highlighted in the limitations section of the manuscript as appropriate

Line 111: Please indicate the time of body composition measurement and the instructions that were given to the participants before the examination (i.e. avoid caffeine, diet on the day of the measurement etc)

Lines 199-203: Please add the percentages of improvement that occur in these parameters after the intervention period.

The authors should explain the positive effect of the nutritional supplement on physical performance and quality of life.

Author Response

  • Title: The authors may consider replacing 'medical food' with ' multicomponent nutritional supplement'

We sincerely thank the reviewer for this and the following suggestions. Accordingly, the title has been modified. (Title)

  • Abstract and Discussion: the authors should avoid the repetition of the content of the examined supplement

Accordingly, the text has been modified avoiding the repetition of the single components of the nutritional supplement used. (Abstract and Discussion)

  • Introduction: Please add a paragraph regarding the negative effect of Covid-19 on functional capacity, body composition, and quality of life of the affected people.

The negative effect of SARS-CoV-2 infection on nutritional status and physical function has already been described in the introduction section. (Page 3, 1st paragraph)

  • Methods: The lack of a placebo-controlled group is one of the main limitations of the study and should be highlighted in the limitations section of the manuscript as appropriate

Even though this issue was just addressed as the first important limitation of the study, in the present revised version of the paper we better describe the type of the study and all the potential correlated biases. Accordingly, new sentences clearly addressing the limitation of the study have been added in the discussion section (limitation of the study). (Page 9, 2nd paragraph)

  • Line 111: Please indicate the time of body composition measurement and the instructions that were given to the participants before the examination (i.e., avoid caffeine, diet on the day of the measurement etc)

Participants were asked to void immediately prior to the BIA assessment since the impedance value is a function of the resistance of electrical current against water flow. No other specific recommendations were given. Accordingly, the text has been modified and new sentences have been added. (Page 4, 4th paragraph)

  • Lines 199-203: Please add the percentages of improvement that occur in these parameters after the intervention period.

All the percentages of improvement that occur in the parameters of interest (hand grip, SMI, chair stand test, walking test, quality of life) after the intervention period are reported in Figure 2.

  • The authors should explain the positive effect of the nutritional supplement on physical performance and quality of life.

In the discussion section all the potential mechanisms that could explain the positive effect of this nutritional supplement on physical performance and quality of life are already addressed.

Round 2

Reviewer 1 Report

The authors have addressed my previous comments to the best of their ability, given the design of the study. Where regression analysis was suggested, it is beyond what might appear in a study such as this, but I would think that the authors would be curious about potential confounding variables within the dataset.

Reviewer 2 Report

No further comments.

Reviewer 3 Report

The authors have addressed my concerns.